# External Support for Elderly Care Social Enterprises in China: A Government-Society-Family Framework of Analysis

**DOI:** 10.3390/ijerph19148244

**Published:** 2022-07-06

**Authors:** Huimin Li, Jianyuan Huang, Jiayun Liu

**Affiliations:** 1Research Institute for Population Science, School of Public Administration, Hohai University, Nanjing 211100, China; 19860034@hhu.edu.cn (J.H.); 180214050002@hhu.edu.cn (J.L.); 2School of Marxism, Taishan University, Tai’an 271021, China

**Keywords:** elderly care, social enterprise, sustainability, external support, government-society-family

## Abstract

As Chinese population ageing becomes increasingly severe, the disjunct between supply and demand for pension services is becoming increasingly serious. The development of elderly care social enterprises plays an important role in solving this disjunction. Such development comes from both the enterprise’s own capacity building and from external support. There are abundant studies on the capacity-building of pension social enterprises in the existing literature, but there are relatively few studies on their external support. In order to better study the external support of elderly care social enterprises in China, we adopted the case study method; we selected GY (a typical elderly care social enterprise in China) as a case study according to certain criteria, and we conducted a series of discussions. Firstly, an analytical ‘government-society-family’ framework was constructed. Second, it was argued that there is insufficient external support for elderly care social enterprises. At the government level, there is a lack of policies, difficulties in implementation and significant geographical differences; at the social level, there are weak support platforms and lack of community supports; and at the family level, there are constraints in regard to traditional concepts and the ability to pay. Finally, an external support system of Chinese elderly care social enterprises was constructed to help more elderly care social enterprises overcome the lack of external support in the development process.

## 1. Introduction

Social enterprises have attracted widespread attention in recent decades [1,2] and have become an effective solution for complex social problems. The literature suggests that the emergence of social enterprises has compensated for the failure of governments, markets [3,4,5,6] and nonprofit organizations to solve social problems and provide public services [7]. Social enterprises are based on financial self-sufficiency, and on the other hand, they seek to solve social inequality and social problems [8,9,10]. The integration of social and economic benefits by such innovative enterprises has attracted attention worldwide [11], and social enterprises have subsequently become an important topic of interest for scholars from different disciplines, such as economics, management, psychology, and sociology. Social inequality and the wide range of fields and populations involved in social problems have led to great heterogeneity among social enterprises. In China, for example, social enterprise services cover 16 social areas, including environmental protection, accessibility services, community development, public finance, pension, education, employment of vulnerable groups, agriculture, poverty alleviation, the Internet, public security, women‘s rights and interests, and focus on 14 specific groups. There are great differences among the social enterprise services available in different social fields and groups. Social enterprises show diversity in their operations and there are great differences among them [12,13]. Therefore, it is meaningful to study social enterprises in different industries [14].

In China, with the increase of population aging, elderly care faces a series of challenges. The sector’s single dependence on the government has been unable to meet the growing demand for pension services, and the enthusiasm of social capital to enter the pension service industry needs to be improved [15]. The overall pension supply has shown a low level of involution [16]. As the main body of pension supply, pension-oriented social enterprises are still in the exploratory stage and face difficulties such as concepts, laws, practices, and talents [17]; thus, they lack the impetus for sustainable development. In this context, we focus on social enterprises in the field of elderly care, which are referred to as “elderly care social enterprises” in this paper. The issue of greatest concern in studies on elderly care social enterprises is sustainability because no matter what kind of elderly care services are provided, only the continuous provision of such services can fundamentally solve the problem of the insufficient rigidity of demand for elderly care. If a change is phased and temporary, it will cure only the symptoms and not the root cause.

Many studies have been conducted on the sustainable development of social enterprises [18,19,20]. However, most of them have focused on the internal capacity building of social enterprises themselves, such as the financial independence [21], legitimacy [19], and competitiveness [22]. Fewer studies have been conducted from the perspective of external support of organizations. The sustainable development of any organization cannot be achieved without external support; therefore, our research focuses on the external support of social enterprises. The individual welfare and public welfare characteristics of the elderly care field make the external support needed by social enterprises different from that of other fields. In this study, we introduce welfare pluralism theory and social support theory and construct a three-dimensional government-society-family framework to analyze external support for the sustainable development of Chinese elderly care social enterprises.

This research makes three contributions to the literature. First, it constructs a government-society-family analytical framework that provides new insight for future elderly care social entrepreneurs into creating new enterprises or promoting the sustainable development of enterprises. Second, as China is experiencing serious population aging, experience in the development of elderly care social enterprises can provide a reference for other developing countries facing the same difficulties or problems. Finally, external support research on elderly care social enterprises can provide a reference for social enterprises in other industries. Only by paying attention to external support and adjusting internal strategies in a timely manner can the sustainable development of social enterprises be achieved.

## 2. Literature Review

### 2.1. Research on Social Enterprise Support

A social enterprise is a business activity that is driven by a social mission [23,24], which does not mean that the enterprise itself is sustainable; it needs to maintain its development in a certain way. This approach can be seen as both the ability of social enterprises to engage in self-reliance (an internal support) and various forms of external support. The first aspect currently garners more academic attention as it consists of more aspects, including leadership, stakeholders, resilient development, sharing practice, brand equity, finance, market competitiveness, social entrepreneurship, and other factors [6,18,25,26]. Social enterprises emphasize the market competitiveness of products and services. For example, the overall goal of BOSKE (i.e., the BOSKE Bakery Cafe, which is in Taiwan and is a social services organization that provides community living and vocational rehabilitation services to adults with intellectual and developmental disabilities) is for people to buy its products because they enjoy its baked goods and services, rather than because they feel pity, compassion, or responsibility toward the individuals [27].

The second aspect is relatively weak regarding garnering people’s attention. Funding sources have a significant impact on the survival and development of social enterprises, thereby making them more dependent on government grants and donor funds, especially in the early stages of creation [28]. Institutional constraints make social enterprises rely more on commercial income during their development [24,29]. This makes it extremely easy for a social enterprise to engage in goal task drift and become faced with the dilemma of social enterprise dual goals [30]. Some scholars believe that infrastructure is an important obstacle to the survival and development of social enterprises [31]. Faced with these obstacles, the local government’s support to social enterprises is very effective [32]. However, when the government provides support, it is temporary and lacks a comprehensive method guidance. This makes such support inefficient and wasteful [33].

Jian, et al. used a content analysis approach to study health organizations seeking social support, including informational, emotional, and instrumental support, through social networking sites [34]. In Berlin, IQ Consulting is a social innovation agency and think tank that supports social enterprise projects with services such as planning, coordination, training, and evaluation [35]. IKure was founded in 2010 with incubation support from the Indian Institute of Technology Kharagpur and Webel Venture Fund, an early-stage incubator established by the West Bengal Government Corporation [36]. Upeeffect was founded in 2016 by social entrepreneurs Sheeza Shah and Sharjeel Chaudhry and is based in London; it currently serves social entrepreneurs in 14 countries, and its crowdfunding mentoring model has resulted in a 95% campaign success rate for social entrepreneurs [37]. Upeeffect’s CEO Shah attributed the low success rate of social enterprise campaigns to a lack of campaign support. The Frugal Innovation Lab at Santa Clara University helps social enterprises by using existing tools [38] to achieve a balance between their social mission and financial aspects. Adequate financial support is an important aspect for the sustainability of social enterprises [39], and the lack of financing is one of the main constraints faced by entrepreneurs when starting a business [40], as shown by the studies of Atieno, Herrington, et al.; Maas and Herrington reported that in the absence of financial support, entrepreneurs cannot predict the survival and growth of their projects [20,41,42], and Casson pointed out that inadequate capital structure and lack of financial resources are the main reasons for the unsustainability of social enterprises [43]. Martin and Eisenhardt used resource-based theory to argue for the financing needs of entrepreneurs, arguing that entrepreneurs need resources such as fixed assets and working capital to gain a competitive advantage in the market [44].

In short, the existing literature on social enterprise support research is still quite weak, because it mainly focuses on how to maintain the balance between a social mission and its financing with single support, such as local government support, i.e., the research lacks a comprehensive discussion of support systems. We believe that it is not enough that the support needed for the sustainable development of social enterprises only comes from financial support. It also needs more support from other sources, and it is better to form a support system.

### 2.2. External Support Analysis Framework: “Government-Society-Family”

Through a literature review, it is found that the existing research on the support of social enterprises has been focused on the specific field of elderly care social enterprises. Furthermore, there is a lack of unified analysis framework in the research process. Therefore, to better study the external support of elderly care social enterprises, we used welfare diversity and social support theory to build an analytical government-society-family framework of external support for elderly care social enterprises.

#### 2.2.1. Welfare Diversification Theory

After the 1980s, many countries around the world experienced a policy shift in the provision of basic services. In many developing countries, the private sector, including civil society- and community-based organizations, has been recognized as part of the formal welfare system [45]. The term “welfare pluralism” was first used in the reassessment of the voluntary sector to describe or explain the process of market and voluntary sector penetration into state functions [46]. This was an important outcome of the implementation of welfare pluralism theory in the social service sector and an important opportunity for social enterprises to come into view in China from the 1990s onward.

Welfare diversification can be seen not only as a way of maintaining or supporting pluralistic systems but also as a way of at least partially replacing the state as a service provider. Welfare pluralism refers to social and health care being provided by four different sectors: the statutory, voluntary, commercial, and informal sectors. The state plays an important role in welfare provision but does not have a monopoly on welfare, which should be provided by the whole society (including the family, the market, and the state) [47]. Olsson, et al. proposed a triadic state-market-civil society model (families, neighborhoods, voluntary organizations, etc.) [48]. Johnson proposed a quadratic model of the state sector-business sector-volunteer sector-informal sector [49], in which the voluntary sector is a concept where the state, the market and the community intersect, with not only voluntary organizations formed by individuals, but also administrative contexts. The voluntary sector is the intersection of the state, the market and the community. Nygren and Pestoff noted that the welfare triangle incorporates the family into the community [50]; however, critics in the family sociology field have remarked that Pestoff ignored the specificity of the family and proposed a quadratic government-civil-associative-private model corresponding to the state sector-market sector-volunteer sector or a third sector—family [51]. In short, welfare pluralism means that social welfare can be provided by different entities [52], whether the state or local government, market, society, community, voluntary organizations, or family. All these entities must undertake certain responsibilities, but the specific systems and cultural environments of each country are different, and the roles or responsibilities that these subjects play therefore differ as well.

#### 2.2.2. Social Support Theory

Research on social support was first proposed in the field of psychology. Durkheim used social support to investigate the physical and mental health of individuals. Later, social support was introduced into the field of psychiatry. In both areas, the main concern was the relationship between social support and physical illness, psychological disorders, mental illness, quality of life, etc. [53,54,55,56]. With the continuous expansion of social support research, social support applications gradually expanded to the fields of social work [57], sociology [58], management [59], etc. In the early days, social support was defined as the exchange of perceived resources between two individuals, the provider, and the recipient, with the aim of enhancing the recipient’s health [60] or a resource provided by another person [61]. Subsequently, some researchers have viewed social support as a complex flow of resources across a support network, not just between two individuals. For example, the Community Support Program (CSP) launched in the United States was aimed at establishing social networks for patients with mental illness in the community, encouraging them to actively participate in social activities and master relevant skills to participate in social activities and helping them truly return to the community. Social support theory emphasizes the social support that individuals receive in their social networks, how that support is provided, how resources flow, and how support networks are constructed.

Clark et al. considered social support to be the help provided by family members, relatives, friends, and neighbors [62], and Wortley W. S. considered it to be the resources available to support individuals when they are in crisis, including other individuals, community organizations, and groups [63]. Social support usually refers to the availability of components of formal and informal support in interpersonal relationships [64]. The objects of social support have evolved from individuals, such as adults, children, older adults, cancer patients, medical professionals, and AIDS patients, to specific groups, such as psychiatric groups, adolescent groups, and even organizations [34], including health institutions, elderly care facilities, and social enterprises. Currently, there is no consensus on the concept of social support in academic circles [65,66,67]. However, social support theory has expanded beyond the original research field, and many scholars have extended the original understanding of the concept and its application. The diverse nature and broad impact of social support have made it one of the most popular constructs in organizational and psychological research [68].

#### 2.2.3. Analytical Framework of External Support: The Government-Society-Family Construct

The theory of welfare pluralism comes from the reform of the European welfare system. It is generally recognized as containing are two important concepts: decentralization and participation [69]. Decentralization refers to the provision of social welfare by the government, the market, social nonprofit organizations, communities, and families, with the government playing the role of arbiter, manager, guide, and coordinator. The essence of participation is to minimize the restriction of nongovernmental participation in welfare supply so that both nongovernmental organizations and those with a demand for welfare can participate in the decision-making process of welfare service provision or planning. These two concepts theoretically provide an important basis for nonprofit organizations to participate in welfare services and bridge the gap between government agencies and individual needs. Compared with Europe and the United States, China has a vastly different traditional culture, politics, and economic system. Therefore, China’s welfare structure has also had its own unique development course. Traditionally, the family has been the main supporter of elderly care, and from the founding of the PRC to the reform and opening-up period, the main supporters of elderly care were members of the family unit. From the reform and opening-up until 2000, because of the emphasis on the market economy, elderly care gradually became the responsibility of the government and the market. Since 2000, as China’s aging population has continued to grow, elderly care is no longer the responsibility of one or two parties but is undergoing a trend of diversification, with the government, the market, society, families, and individuals all becoming responsible for elderly care, which also reflects the idea of welfare diversification. In the 1980s and 1990s, elderly care social enterprises gradually emerged. As a new form of social organization and one of the main providers of social services, these enterprises have made great contributions to solving the social problems of the aging population in the field of social welfare.

Elderly care social enterprises are gradually taking their place as hybrid organizations in the field of social welfare. Sud et al. argued that social enterprises, like every new type of organization, must be accepted politically and socially. The existence of certain types of organizations depends on the support of the society in which they are located [70]. Studies by Defourny, Nyssens and Kerlin showed that country-specific conditions lead to different institutional and legal environments [71,72,73]. These external support environments determine the legitimacy and extent of the development of social enterprises as part of the social sector. The stronger the external support an organization has, the more it can integrate different resources to achieve sustainable development in a constantly changing environment. At present, there is no perfect theory for the external support of elderly care social enterprises. Therefore, we examine social support at the organizational level [64] and construct the external support system of elderly care social enterprises by applying the theory of social support and social welfare diversification.

The previous review of social support and welfare pluralism theory shows that social support includes a support subject and a support object. The support subject can be a person’s informal network, such as family, friends, and colleagues, or formal network, such as health care professionals and human service workers [74], and support can be provided by different types of organizations, such as government agencies, NGOs, communities, social networks, and social work experts. The support object can be an individual, a group, or an organization. The subjects of social welfare provision can appear once or more often depending on the historical period, country, culture, and even system. The subject of social welfare provision can play the role of both the support subject and the support object. Previous research has not attempted to combine the two theories. This study does so and builds an external support analysis framework for elderly care social enterprises according to the actual situation of elderly care in China. This definition extends the object of social support from conscious individuals or groups to the field of organization in a further exploration of social support theory. Support subjects can be provided by other subjects of social welfare, such as government agencies, society, family members and elderly individuals, to constitute the external support system for the sustainable development of elderly care social enterprises, which is also the government-society-family analysis framework (Table 1). According to the actual situation of China, the government in this framework refers mainly to the national and local governments at all levels, and the support methods that they adopt are mainly policy support, financial support, and legal support. Society refers to various organizations other than the government and family, such as enterprises and institutions, nonprofit organizations, and autonomous organizations, which mainly provide certain kinds of organizational support and community platforms for elderly care-oriented social enterprises. Family includes relatives of elderly individuals, who maintain the sustainable development of elderly care social enterprises mainly through changes in traditional elderly care concepts, economic support, and spiritual comfort. The purpose of this study was to analyze the actual situation of the external support for elderly care social enterprises in China from the perspective of external support analysis framework, and to construct an external support system of Elderly care social enterprises to promote the sustainable development of elderly care social enterprises.

## 3. Methodology

### 3.1. Case Study Methodology

For this article, we adopted a qualitative case study approach and selected a potential elderly care social enterprise (“potential social enterprises” can also be understood as “quasi-social enterprises”, i.e., a phenomenon that occurs in the current environment of China’s social enterprise certification system, where there are physical organizations that are not officially recognized), which is called GY in this paper. Starting from the three-dimensional government-society-family framework, this paper explored external constraints on the development of China’s geriatric social enterprises and at the same time aimed to identify relevant external support to ensure the sustainable development of geriatric social enterprises. In other words, we investigated what kinds of factors constrain sustainable development and how to achieve sustainable development through external support for enterprises. The question of how is well suited to the case study approach [75]. In the case study process, we conducted participatory observation by examining specific GY scenarios and actually participating in the daily social life of the research subjects; on the other hand, we conducted in-depth interviews with relevant people, such as the founder of GY, the main person in charge, the caregivers, the elderly individuals, and family members of the elderly individuals to determine their views on elderly care social enterprises and on elderly care social enterprises. The purpose of this study was to grasp the views, attitudes, emotions, and behaviors of those involved with GY to obtain firsthand information for this study. In addition, we conducted in-depth interviews with professional authorities to obtain more information about the sustainability of elderly care social enterprises to compensate for the lack of written information and details of typical social enterprises. At the same time, we performed non-participatory observation of other similar social enterprises to investigate their operations, the actual feelings of the elderly individuals and the sustainable development of the enterprises, thus creating a supplement for the data of typical cases. By collecting data from professional authorities and other social enterprises involved in elderly care services, we further confirmed the typicality of the GY in this type of social enterprises. We spent six months participating in observation and field interviews and recorded first-hand information in real time. In general, the triangulation method in qualitative research strategy was adopted, including the methodological triangulation method and the researcher triangulation method. This approach allowed us to obtain a more comprehensive understanding of the research phenomenon. According to the academic research norms, the names of organizations and personnel in the text have been technically processed (Table 2).

### 3.2. Case Selection Criteria

First, there are certain recognized characteristics of a social enterprise. Although there is no uniform definition of a social enterprise, different countries have different understandings of social enterprises. However, generally speaking, social enterprises are seen both as a powerful complement to government, market and nonprofit organization failures in some areas, and as businesses that use business models to solve social problems and create social value, while maintaining a strong sense of social mission and social responsibility.

Second, the social enterprises examined in the current study provide elderly care services; thus, other types of social enterprises are not within the scope of this study. There are many fields of social enterprise services, such as those pertaining to children and youth, barrier-free services, rural development, ecological protection, elderly care and so on. Given that there are differences between the different fields of social enterprises, we only focus on the field of elderly care of social enterprises. These enterprises will be collectively referred to as elderly care-oriented social enterprises. Their service objects are mainly elderly individuals and their families, and their function is mainly to provide various long-term or short-term old-age care services, such as life care, medical care, spiritual comfort, emergency relief, breathing services, home services, and so on.

Third, there in the problem of time malleability. Time span is a very important factor in the sustainable development of social enterprises. The GY is a representative of the three types of social enterprises for the elderly population that meet the above mentioned three criteria.

With the help of the Civil Affairs Bureau, we got a list of 12 better elderly care enterprises. Then, according to the above three standards, we finally chose GY. According to Standard 1, two enterprises were excluded; according to Standard 2, three enterprises were excluded; according to Standard 3, six enterprises were excluded.

### 3.3. Case Profile

The GY was founded in 1999 and has been developing for more than 20 years. In 1999 the government began to allow social capital to enter the field of elderly care. The first private elderly care institutions in our country were established in 1999; the current Chinese population ageing problem also began in this year. Prior to 1999, care services for the elderly population were basically carried out by the children in a family; thus, the demand for socialization of care services for the elderly population was not strong. At this point, China’s elderly care services did not cause widespread concern within the community or serve as a social problem. As the country entered the current year of population ageing, families began to increasingly struggle to provide care for their elderly relatives. The rigid demand of China’s elderly care services was seriously insufficient, which caused widespread concern from all walks of life. In this context, the elderly care industry began to develop.

In its initial period, the GY was a simple private enterprise for the aged, which mainly provided life care service for the aged with disabilities and mental handicaps and cooperated with hospitals to provide medical rehabilitation service. During this period, the GY had 60 beds and offered the concept of integrated care. At that time, due to the lack of corresponding policy system and the support of relevant departments, the implementation of integrated care concept was very difficult. In 2003, the GY was transformed into a public-funded private pension enterprise that was supported by government infrastructure; the number of beds increased to 106.

In 2011, under the guidance of national policy, the GY began to explore the chain operation model, and successfully created a second retirement home. Thus far, there are 12 chain organizations. The chain’s geographical area covers the southern, northern, and central areas of Jiangsu Province. At the time of the opening of the second institution, the GY put forward its social value concept of providing high-quality services for elderly individuals at low cost; these services included personal health care, diet and personal care, home health care and social and recreational activities. The GY also regularly participated in the community to carry out free health check-up activities, thereby going as far as possible to solve the problem of unattended elderly individuals. Thus, the GY had begun to pursue the dual goals of social enterprise.

However, in China at that time, the perception of social enterprise was very vague. On 3 June 2011, the ninth plenary session of the 10th Beijing Municipal Committee of the Communist Party of China adopted the “Opinions of the Beijing Municipal Committee of the Communist Party of China on Strengthening and Innovating Social Management to Promote Social Construction in an All-round Way”, the third part of which encourages localities to “actively support the development of social enterprises and vigorously develop social service industry”. This was the first mention of social enterprises in official documents. On 11 November 2016, the General Office of the Communist Party of China Beijing Municipal Committee Beijing Municipal People’ s Government issued “Beijing’s 13th Five-Year period of social governance planning”, in which the third part of the fourth article clearly points out that one should “vigorously promote the development of social enterprises focusing on serving the people’s livelihood and carrying out public welfare”. On 23 April 2018, the Chengdu Municipal Government promulgated the “Opinion on Cultivating Social Enterprises for Community Development and Governance”, which was the first relevant policy specifically targeting social enterprises. Thus far, this is the only document specifically aimed at social enterprises; there is no similar national document or guidance at the specific practical level.

There are many geriatric care social enterprises, or potential social enterprises, in the field of practice that are like the GY. The main reason for calling the GY a potential social enterprise is that the status of their social enterprise is not clear. Although the GY has been practicing the dual goals of social enterprises, they have no clear identity at the legal level, which results in its development process being subject to many restrictions. Initially, the GY was positioned as a private elderly care institution. Later, it cooperated with the government and became a public-private enterprise that was supported by the government. At present, it is positioned as an elderly care social enterprise. However, there are some controversies over elderly care social enterprises in both academic and practice fields, and the dual objectives of social enterprises (social and economic benefits) are like a double-edged sword because these enterprises are considered both a model innovation and a source of controversy. In short, social enterprises started late in China and are still in the exploratory stage. Many private nongovernment enterprises, institutions or organizations already have various elements or characteristics of social enterprises but do not have the “self-awareness” of social enterprises. Many social enterprises are torn between presenting themselves as general commercial enterprises or nonprofit organizations for the public good, and they do not have a clear understanding of their own function. “When I first came to GY, I thought it was a nonprofit social organization but later found that GY was in the development process, and except for government-related preferential policies, its development basically relies on its own profitability. However, there are a lot of elderly care institutions in society; no matter public, private or public-private construction, their identities are difficult to accurately position, so the positioning of GY is really difficult to say” (Interview record: 20201018).

## 4. Results

At present, the GY has achieved a certain degree of development; however, it still faces a shortage of external support in its development process, and these shortcomings have restricted its development to a certain extent. Through participant observation of the GY’s development process, the coding and analysis of relevant interview materials and observation notes and combining the external support analysis with the government-society-family framework, we found that the external support deficiencies affecting the sustainable development of elderly care social enterprises are reflected in main three levels: government, society and family.

### 4.1. Government: Difficulties in Policy Implementation and Significant Geographical Differences

#### 4.1.1. Weak Policies and Difficulties in Landing during the Start-Up Period

Government support was lacking during the start-up period of the GY. Analysis of the GY interview data revealed that this lack of government support can be understood in two ways.

First, there was a lack of relevant policy support. The Ministry of Civil Affairs of the People’s Republic of China issued the Interim Measures for the Management of Social Welfare Institutions at the end of 1999, which was the only governmental policy document at that time, and this document addressed only the macrolevel aspects of liberalizing strict control over elderly care service providers and allowing social forces (social organizations, enterprises, communities, individuals, etc.) to participate in elderly care services. Regarding what kinds of support the government would provide and what elderly care institutions should do, there is no clear leading document, and it is exceptionally difficult to create elderly care service institutions. The founder of the GY has strong feelings in this regard: “During the start-up period, the founders borrowed the start-up capital, rented an unused room, and ran around Nanjing’s old people’s homes, orphanages, and senior apartments to gain experience in elderly care services. It took nine months to go through the approval process. There was no support from the national policy system in this process” (Interview record: 20191018). Regarding the process of setting up the first elderly care service organization, a staff member from the management department said, “When your legs are an inch thinner, the elderly care service organization will be successful”. This means that the procedure to successfully register an elderly care social enterprise is very cumbersome, i.e., managers need to go to many departments in different regions or different functions to complete the necessary procedures (Interview record: 20191023). The procedure was extremely cumbersome and took nine months. The first elderly care service agency moved three times.

Second, the policy system was difficult to implement because of poor coordination due to geographical and population divisions, the “one-size-fits-all” approach, and a lack of detailed provisions in land, finance, taxation, and other preferential policies, resulting in a lack of supporting policies for elderly care service institutions. In actual operations, it was also difficult to implement operating subsidies and caregiver subsidies for some elderly care institutions. The participants observed that the government had a preferential policy for elderly care service institutions in applying for discounted loans from The China Development Bank, but the bank does not provide such loans. Additionally, it was difficult to apply for certain funds allocated to elderly care institutions, such as the provincial pension industry fund jointly established by the Jiangsu Provincial Finance Department and Jinling Hotel Group. Therefore, insufficient financial support has limited GY’s pace of development to a certain extent (Interview record: 20191020).

#### 4.1.2. Serious Policy Fragmentation of GY during the Trial Period (GY Has a Heavily Divided Policy for the Continuous Trial Period)

Along with the aging of the population in China, elderly care services have attracted increasing attention in all walks of life. The government has introduced a series of policies to realize the rational allocation of resources and meet the demand for elderly care services, and the GY has also entered the development chain period with the prevailing trend of government policies. However, during this period, the GY encountered serious divisions in the chain development process due to regional and population differences in support policies for elderly care. This division hindered the chain process and the sustainable development of elderly care social enterprises.

First, in terms of construction subsidies, operational subsidies and care subsidies, there were differences among cities and regions, and some of these differences were quite large. Table 3 shows differences in construction subsidies among different regions; the construction subsidy in Suzhou in southern Jiangsu was 20,000 yuan, while the construction subsidy in Xuzhou in northern Jiangsu was only half that amount. Table 4 shows that the monthly subsidy for disabled elderly persons (city household registration) was 300 yuan in Nanjing, while it was only 60 yuan in the Nantong region. Table 5 shows that the daily care subsidy for severely disabled elderly individuals was 70 yuan in Nantong, 26 yuanin Suzhou, and 30 yuan in Xuzhou. Regarding per capita household pension income level (Table 6), there were also large gaps between cities; for example, the per capita household pension was 7870 yuan in Suzhou, while it was only 2905 yuan in the Lianyungang area.

Second, the gap between urban and rural areas of the same cities was also obvious. For example, in Suzhou, the per capita household pension for urban residents was 9534 yuan, while that for rural residents was only 4129 yuan (Table 6). Figure 1 shows that the per capita pension of urban and rural residents in the urban area of a city was 259 yuan/month, that of retired staff of enterprises was 2614 yuan/month, and that of institutional retired persons was 7326 yuan/month. These differences among regions and populations have seriously hindered the chain development and replication of elderly care services. The chain development process of the GY has been affected by these policy divisions, which have led to different development trends in southern, northern, and central Jiangsu. The development trend in southern Jiangsu has been better than that in central and northern Jiangsu, while development in northern Jiangsu has been difficult. Both the quality of elderly care services and the sustainability of the GY chain have been unsatisfactory.

### 4.2. Social Level: Weakness of Support Platform, Lack of Community Support

#### 4.2.1. Weakness of Organizational Support Platforms: Difficulties in Private Certification of Social Enterprises

Support organizations for social enterprises in the private sector recognize the importance of identifying them as social enterprises, but support for social enterprises remains weak.

It was not until 2015 that the first civil social enterprise certification scheme emerged in China, namely, the China Charity Fair Social Enterprise Certification Scheme (Trial) issued by China Charity Fair Shenzhen. In 2015, only seven enterprises or social organizations in mainland China were recognized as social enterprises; the number increased to 16 in 2016 and 23 in 2017. In 2018, the Chengdu Social Enterprise Certification Center was established to create a comprehensive service platform for social enterprises in Chengdu, and in 2019, 46 enterprises and social organizations were certified as social enterprises in Beijing. As of January 2020, the number of social enterprises that had completed certification was 283. At present, few social organizations and enterprises have been certified as social enterprises, and more are considered quasi-social enterprises or potential social enterprises. Social enterprises have been operating in OECD (Organization for Economic Co-operation and Development) countries for almost 30 years, and as of 2015, there were 70,000 social enterprises in the UK (The United Kingdom of Great Britain and Northern Ireland), over 50,000 in Spain, and over 5000 in Scotland. Social enterprises have become the epitome of social innovation and the main force in the field of public welfare. At present, 22 countries have relatively complete social enterprise certification systems, and some European and North American countries have multiple certification models.

#### 4.2.2. Community Level: Low Social Acceptance and Insufficient Participation

Elderly care social enterprises can obtain government resources through government purchases of services and provide care services for elderly community members. For example, the GY provides free blood pressure and blood glucose measurements and related health consultation services for elderly individuals in the surrounding communities as well as meal purchasing services, rehabilitation services and nursing care services. However, in the process of chain location, the GY has encountered a serious “neighbor avoidance effect”; its locations have caused strong dissatisfaction among residents in surrounding communities who opposed the construction of elderly care institutions or senior service centers and opposed any form of day care or full care, creating many obstacles for the GY in its chain development process. Through detailed interviews, we found that residents of surrounding neighborhoods were worried that because elderly individuals are more prone to illness, more sick people would increase medical waste, which would not be conducive to the neighborhood environment. Additionally, elderly residents would be concentrated in one place, which would give neighbors an uncomfortable feeling of twilight and consume more public resources. Finally, with more elderly residents, there would be more funerals, which the neighbors did not want to see. On the other hand, in communities with more young residents, neighbors thought it would not be very meaningful to establish an elderly care community enterprise, and they believed it would reduce the quality of life and housing prices in the community. “If it is the government’s planning that requires the establishment of related senior care services to facilitate the needs of the elderly population, we have nothing to say about this kind of thing, but this kind of senior care social enterprise is for-profit, commercial, and occupies the public resources of the community, which directly affects the rights of other people in the community, which is unreasonable” (Interview record: 20191118). Therefore, the social acceptance of elderly care social enterprises in the community is not high. This situation is due to the late start of social enterprises in China and the lack of publicity to attract the attention and improve the understanding of the general public. On the other hand, elderly care social enterprises have failed to fully explore the strength of the community in the development process, resulting in insufficient community participation and lack of joint efforts to solve the problems of elderly care in the community. A community may have natural advantages in providing elderly care services, such as certain sites, human resources, community purchases of elderly care services, and active participation of community residents. Therefore, the GY did not fully mobilize the community and its residents’ motivation in the actual operational process, resulting in low social acceptance and insufficient community participation.

### 4.3. Family Level: Traditional Concept of Bondage, Payment Capacity Constraints

Among elderly people, there is a certain prejudice against elderly care social enterprises, and they prefer to receive care from their children or professional medical services in hospitals rather than from elderly care social enterprises. Their children are afraid of appearing “unfilial” or fear that they will be stigmatized if they send elderly family members to elderly care social enterprises. Due to the traditional concept, some families have even experienced tragic accidents and deaths caused by a lack of caregivers and a lack of timely detection of health problems. Modern changes in the culture of filial piety have come, but the whole is still locked in the shackles of the tradition, and some old people will worry about their children being condemned by public opinion and do not want to accept the services provided by the elderly care social enterprises.

In our study of the GY, we heard a story about Grandpa Lu from interviewee W7. Grandpa Lu, 73, who had worked as a surgeon in his youth, suffered from high blood pressure and diabetes for years and gradually showed signs of brain atrophy. He could not take care of himself and often left the house on his own. However, at that time, the decision of his two children to move him to a GY facility was controversial. Their mother was the first to disapprove, and her relatives, including her uncle, questioned why Grandpa Lu had been sent there. Colleagues, friends, and neighbors also regarded their behavior as unfilial. However, they worried that their father would get lost, that their mother would be dragged down one day, and that they would lose their father because they could not care for him properly. The public opinion of the outside world, their own ambivalence, and the criticism of colleagues led the two sisters to tears of sadness when recalling their experience (Interview record: 20191130).

Some elderly individuals have limited ability to pay, and elderly care social enterprises must meet the demand by providing relevant services that require a certain expenditure. Although such costs are lower than those of for-profit elderly care institutions, they still require economic input from elderly residents and their families. According to the results of the survey, elderly individuals and their families have not formed a sense of consumption of elderly care services. Except for the home-based services supported by elderly care subsidies, families are less willing to purchase elderly care services at their own expense, and they are too dependent on the government. “If it is a free service, I will use it, but if I need to spend my own money to buy this, it still needs to be carefully considered, especially what spiritual comfort aspects, psychological aspects of these feel less useful, and their pension cannot pay” (Interview record: 20190930). The reasons for this limited ability to pay, from the perspective of development, include the relatively low level of economic development and the imperfect social security system. At the regional level, there is still a large gap between northern Jiangsu, with relatively low economic development, and southern Jiangsu, with high economic development, and the pension level in northern Jiangsu is generally low. Therefore, many elderly people with service needs cannot purchase elderly care services due to their limited ability to pay, which also hinders the sustainable development of elderly care social enterprises.

## 5. Discussions

Although the GY shows a good development trend at present, it is difficult for one or a few potential social enterprises to solve the social situation of insufficient demand for elderly care services in general. Moreover, as the previous analysis shows, the development process of the GY has been severely constrained by the lack of external support, which affects its sustainable development. This lack of external support restricts the creation and development of more social enterprises, resulting in a lack of self-awareness in many potential elderly care social enterprises, which makes already created social enterprises unsustainable or even ultimately causes them to disappear. Therefore, based on the in-depth analysis of the lack of external support for the GY, we constructed an external support system for elderly care social enterprises to create a friendly social environment in order to promote the sustainable development of such enterprises as an important main body of social welfare provision in China. The support of government, society, and family, as relevant stakeholders, has a positive contribution to elderly care social enterprises. This is consistent with the work of Phillips et al. [76].

### 5.1. Government Support: Top-Level Guarantee for the Sustainable Development of Elderly Care Social Enterprises

Government support is the top-level guarantee for the sustainable development of elderly care social enterprises and is reflected mainly in supportive and preferential policies, funding sources and recognition of legal status.

First, the most important point from the perspective of supportive and preferential policies is reserving a place for elderly care social enterprises in the design of the social security system. According to the theory of welfare pluralism, the status and role of social enterprises in the provision of social services should be clarified, and a certain amount of funding should be provided for elderly care services. In terms of specific policy implementation, the government should consider the uneven development of elderly care social enterprises caused by population and geographical differences and pay attention to decentralization when formulating specific policies. Additionally, it should give full rights to the functions of local governments so that localities can introduce management policies to support the development of local elderly care social enterprises. For example, the government can implement the policy of “new homes with new methods and old homes with old methods” when issuing fire certificates for elderly care services, and new elderly care social enterprises should strictly follow the policy when applying for fire certificates. Established elderly care social enterprises or those converted in old neighborhoods should pass the fire department safety inspection to ensure that there are no safety hazards as the main criteria for renewing an establishment license. On the other hand, tax incentives and operating subsidies should be provided for social enterprises in terms of specific policies, and social enterprises should be prioritized in implementation, such as in the process of government purchases of services. The government can consider adding the social value factor when selecting procurement objects, examine the economic value-price ratio and social value-price ratio of different service providers, and select social enterprises that have price advantages and can attract good social value feedback. Based on the international development trend, in facing obstacles to the development of social enterprises, it is a trend for governments to promote the development of social enterprises through public policies [77]. The support of central and local governments to social enterprises has been paid more and more attention [26,78]. Institutions constrain the development of social enterprises [24]. In Britain, the current policy has an important impact on social enterprises [79].

Second, funding is a sufficient condition for the creation and sustainable operation of an organization. The public interest and welfare nature of elderly care services make elderly care social enterprises less profitable than general commercial enterprises, which leads to low interest in this industry among market capitalists and a lack of motivation to enter this industry, making it difficult for social enterprises to raise capital and find financing. This requires the government to help social enterprises in terms of funding; for example, the government can provide certain guarantees or assistance when social enterprises take out loans so that these enterprises can obtain lower loan interest rates than general commercial enterprises. Additionally, the government can establish social enterprise foundations and social enterprise start-up funds to provide financial support for entrepreneurs who are willing to create social enterprises. For example, the UK established the first Social Investment Bank in 2012 to provide more comprehensive financial services to social enterprises and established the Social Enterprise Incubator Fund to support social enterprises in their start-up phase. The government can also provide direct financial support to social enterprises to make it less expensive for social enterprises to obtain loans, set up a special loan fund for social enterprises that can be used to subsidize interest rates for small and medium-sized enterprises. The government can also encourage various foundations to set up social enterprise fund programs by providing tax incentives, opening “green channels” and so on. Many international organizations begin their support for social enterprises by providing financial support, such as loans and grants [80]. However, some studies have also pointed out that social enterprises need to reduce their dependence on grants to maintain financial independence [20].

Again, in terms of legal status, the public in China has a high degree of recognition of the government, especially the central government. Although social enterprises have been certified by the private sector, at the government level, social enterprises still do not have their own legal identity, which has led to a certain degree of public skepticism toward social enterprises and low recognition of their social and economic value. It is imperative to legally recognize the status of social enterprises, yet the 2007 Korean Social Enterprise Promotion Act (SEPA) is the first and only social enterprise legislation in Asia. In addition to macrolevel policy and institutional support, government support for social enterprises strengthens the links among resources and clarifies the boundaries and scope of social enterprises. The legitimacy of social enterprises facilitates their access to more social resources [81].

### 5.2. Social Support: Social Environment Guarantee for the Sustainable Development of Elderly Care Social Enterprises

A friendly social environment is essential for the long-term development of elderly care social enterprises. There are many elements of a friendly social environment, and the emphasis here is on the support of organizations and community support. Organizations in various forms are important partners in the development of elderly care social enterprises. The community is a very special place in China. it is an autonomous organization, and it performs many administrative functions; thus, organizational support and community support play an important role in the development of elderly care social enterprises. The Social Impact Incubator is of great significance to the early development of social enterprises and the realization of their mixed goals [2]. Community Empowerment enables social enterprises to access more community resources [31].

The public welfare and microprofit nature of elderly care social enterprises require cooperation with related organizations, and the partners must have a certain sense of social responsibility and not focus on maximizing profits. In our case study, we found that the synergy of organizational support provided a constant impetus for elderly care social enterprises. For example, Golden Aging, the social enterprise that we interviewed, received support from its partners in the creation process. Four-fifths of the funds had come from friends, and the renovation company not only did not call for money for the renovation work but also advanced a certain amount of money that was not paid off until five years after the enterprise was founded. It is a good-hearted company with a sense of social responsibility that to a certain extent relieved the financial pressure at its beginning and provided help hand for its sustainable development. With the help of the Northeast Chamber of Commerce, the GY received the opportunity to participate in a three-person joint guarantee by the Postal Savings Bank and was eligible for a loan for two consecutive years; the Northeast Chamber of Commerce actively contacted various media to publicize and report on the GY to raise public awareness. Additionally, it invited relevant organizations in China and abroad to visit and guide the GY, thus increasing social awareness of the enterprise. The development of the GY also benefited from the support of certain organizations, such as companies that provide products for the elderly population, companies that provide food, and web-based information technology services.

The community is the basic arena for aging in place, which many Chinese seniors are currently seeking to do, as they want to age in their own familiar community environment. Community support is the beginning of social enterprise service provision for elderly care. Elderly care social enterprises have a natural and close connection with the community. Judging from the development trend of elderly care, community care has become the common choice of different countries and regions, whether abroad or in China, and people prefer to receive elderly care services in their familiar living environment. As the main providers of elderly care services, social enterprises can meet people’s needs thus achieve their own sustainable development. The community can provide field support for elderly care social enterprises with the participation of community members, cultivate volunteerism, provide resources for less aged elderly people, and realize active aging. The community can provide venues, facilities, and even financial support. The participation of social enterprises in elderly care services is complementary to the realization of China’s home-based, community-based, and institution-based elderly care service system. For example, in the early stage, the GY set up a care institution for elderly people with disability and dementia as its main service object. In the process of chain development, it continued to cooperate with the community; actively respond to the actual needs of elderly people; take community elderly care as an important breakthrough point in the chain process; seek various kinds of community support; set up community elderly service centers, day care centers, elderly activity centers, and respite services; and increase the provision of medical care, rehabilitation aids, cultural, sports and entertainment activities, and clothing, food and catering services to create the necessary conditions for aging at home. The community is the main site of social participation. Social enterprises should fully utilize community resources and mobilize residents to participate, especially younger and unemployed elderly people, to provide social enterprises with volunteers and identify elderly people who may need care. At the same time, they can promote themselves in the community and improve public acceptance and recognition of social enterprises. Therefore, community support is a resource integration platform for social enterprises to participate in elderly care services. Through this platform, community acceptance and resource integration of elderly care social enterprises can be better realized to improve community participation.

### 5.3. Family Support: The Cornerstone of Sustainable Development of Elderly Care Social Enterprises

Family support is reflected mainly in changes in the family concept of aging, financial support, and spiritual comfort. Without the support of elderly individuals and their families, elderly care social enterprises are equivalent to water without a source and wood without a foundation. Therefore, in the process of social enterprise participation in elderly care services, the actual needs of elderly people and their families should be fully considered to obtain their support.

Filial piety is a traditional cultural practice in China, but the specific forms of filial piety have changed considerably. Traditional forms of filial piety, such as “parents in, do not travel far”, “seeking carp on ice”, “burying children and serving mothers”, and “warming pillows”, have undergone great changes due to the large population flow, changes in forms of family cohabitation and finances, improvements in women’s family status, the decline of traditional patriarchal authority, and changes in intergenerational relations. The traditional concept of raising children to support parents in their old age has been challenged, the importance of daughters in supporting parents in their old age has been gradually recognized, social old age has entered people’s perceptions, and various forms of old age institutions are no longer avoided by elderly people and their families. In short, with changes in people’s concept of family aging, there will be a broader place for elderly care social enterprises.

As China’s economic development continues to increase, per capita disposable income will continue to rise (Figure 2). The increase in disposable income per capita has increased the ability of families to pay for elderly care services. Although not all households can do so, the reality is that some households do have increased ability to provide financial support for elderly individuals [82]. Moreover, changes in family structure (such as “4-2-1” families and empty-nest families) and the lack of professional elderly caregivers (family members lack certain professional service knowledge and skills, making them unable to cope with elderly relatives with disabilities and dementia) have led to the weakening of family elderly care service provision. As people’s life expectancy increases, the expansion and compression of illness and disability and survival with illness have become the norm for the elderly population, and the traditional and purely material aspects of care provision can no longer meet the actual needs of elderly individuals. In short, there are families that have a certain financial ability to pay for elderly care services, but a lack of family members to provide such care or a lack of expertise related to elderly care leads to difficulties. Therefore, to the extent that the family’s financial ability allows, purchasing social services has become a rigid demand for many families, i.e., with further increases in wage levels, children will provide adequate elderly care services for their parents by purchasing elderly care services in the market [83]. This is also a source of motivation for the sustainable development of elderly care social enterprises.

### 5.4. Limitations and Family Support: The Cornerstone of Sustainable Development of Elderly Care Social Enterprises

The sustainable development of elderly care social enterprises requires external support, i.e., exogenous guarantees, which is the main content of our research. On the other hand, elderly care social enterprises need to develop their own internal power sources, i.e., to develop their own advantages, including social entrepreneurship, innovative services and products, branding, chain business models, and advanced social enterprise concepts. Through the integration of “self-blooding” and “external blood transfusion” of social enterprises, the social support network of elderly care social enterprises can be built as a whole, and the social objectives and benign operations of social enterprises participating in elderly care services can be realized to achieve sustainable development. The terms “self-blooding” and “external blood transfusions” are widely used in many fields in China, such as the development of public welfare organizations, poverty alleviation and rural revitalization. The meaning of “self-blooding” means to focus on the internal motivation of things, i.e., through their own efforts to achieve development; “external blood transfusions” refer to external factors, i.e., the development of things through external support. In this process, “self-blooding” and “external blood transfusion” are indispensable, but our research focuses on external support for elderly care social enterprises and does not provide an in-depth analysis of the endogenous dynamics of elderly care social enterprises. Therefore, our study ignores the internal motivation of social enterprises to participate in the sustainable development of elderly care services. In addition, social enterprises, as social organizations, should be placed in a social ecosystem to comprehensively investigate their sustainable development. Only by specifically examining their interactions in the social ecosystem can we comprehensively study their sustainable development, which is the direction of the author’s future efforts.

In addition, there are certain limitations in using a typical case study. One case can represent only one, or one type, of specific situation, and it is difficult to generalize the findings to the complexity and diversity of real elderly care social enterprises. In future research, we hope to supplement the shortcomings of one case study by using a multiple case study approach to include different types of elderly care social enterprises as much as possible.

## 6. Conclusions

The GY as a potential elderly care social enterprise has achieved some sustainable development since its establishment in 1999. Firstly, it solves the care needs of some elderly individuals and continues to meet the care needs of elderly people in more areas through chain development. To some extent, it compensates for the deficiency of the government, the market, nonprofit organizations, and families in providing care services for elderly individuals and realizes the sustainable development of social value. Second, the GY has realized its own economic independence and sustainable development while realizing its social value. It depends on the preferential policy support of the government, but not entirely. Through the business operation mode and chain brand effect, personalized services are provided for elderly individuals with different needs, which leverages the consumption of care services by elderly people and their families and realizes the sustainable development of economic value.

In the early days of the establishment of the GY, people’s assessment of the needs of elderly care services was unreasonable, and the preparation for dealing with population aging was insufficient. Although the government has relaxed the policy for elderly care service providers, especially for private elderly care service providers, there is no specific policy system for elderly care service providers at the highest level of the government. The GY‘s chain of development has been a period of intensive government support. They receive much support from policy to funding, from the central government to local governments. The government’s support for elderly care-oriented social enterprises is very important, and some studies have shown that these support policies have effectively promoted their prosperity and development. However, in the initial stage and development stage of these enterprises, due to the temporary and biased support, their effectiveness is insufficient. Therefore, in the process of policy support, it is necessary to evaluate and monitor the policy for a long time.

Until now, the development of social enterprises in China has been very slow, and China has no clear official regulations regarding the identity of social enterprises. However, many potential social enterprises have been active in youth education, elderly service, disabled services, community development, rural poverty alleviation, public security and other fields, and they have achieved many positive results. The government gives a clear legal identity to elderly care social enterprises, which can improve public recognition of social enterprises and enhance their credibility. For example, the UK introduced the Public Services (Social Value) Act in 2013, proposing that the government should examine both economic and social value factors when purchasing public services and prioritize purchasing services provided by social enterprises when other circumstances are equal. In Korea, the Social Enterprise Incubation Act requires the government to submit a plan for the purchase of social enterprise products and the number of purchases to the Ministry of Labor each year.

Community plays an increasingly prominent role in the lives of the elderly population. When elderly individuals return from previously held jobs to the community, the community becomes the focus of their lives. Therefore, on-site elderly care has become the choice of many elderly people, which is also related to the actual needs of their families. Thus, elderly care social enterprises will engage in more sustainable development if they want to get full support from the community.

The traditional concept of old-age care has a history of thousands of years in China. With the advancement of modernization, the traditional concept of old-age care has undergone new changes, but it is still deeply rooted in the overall culture. The family pension service is still the first choice for most people. This factor makes them hesitate to choose elderly care social enterprises.

The change in the concept of family aging and the increase in family economic support for the elderly population does not mean a weakening of the family’s function in the aging process but rather a wider acceptance of the services provided by elderly care social enterprises. The enterprise itself should have the ability to meet the actual needs of elderly people; family members should make elderly people feel the warmth of their children while receiving these socialized services. In a country such as China, where filial piety culture is a long-standing tradition, the spiritual comfort of elderly people (the joy of having grandchildren, the happiness of family and grandchildren) is extremely important and cannot be provided or replicated by any elderly care social enterprise outside the family. Therefore, to achieve the continuous development of elderly care social enterprises, the provision of spiritual comfort to elderly individuals by family members is essential. While elderly individuals receive elderly care services provided by social enterprises, family members should take responsibility for spiritual comfort through regular visits by the younger generation, reunions on traditional holidays, and video calls using modern internet tools. Only through this two-pronged approach can elderly individuals feel the warmth of their families while receiving socialized elderly care services, which will improve their quality of life. Our research shows that government, society, and family play an important role in the development of social enterprises for elderly care.

## Figures and Tables

**Figure 1 ijerph-19-08244-f001:**
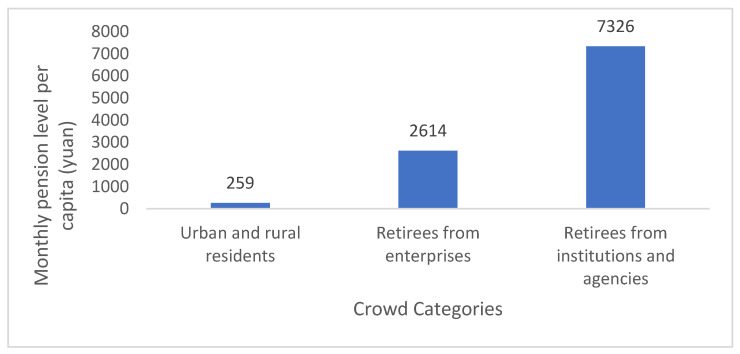
Per capita pension levels for different population groups in a city’s urban area, 2021. Source of data: Compiled from relevant notices and announcements of the Human Resources and Social Security Bureau of the city. Note: The main targets of urban and rural residents’ basic pension insurance are urban residents who do not participate in the basic pension insurance for enterprise employees as well as rural residents.

**Figure 2 ijerph-19-08244-f002:**
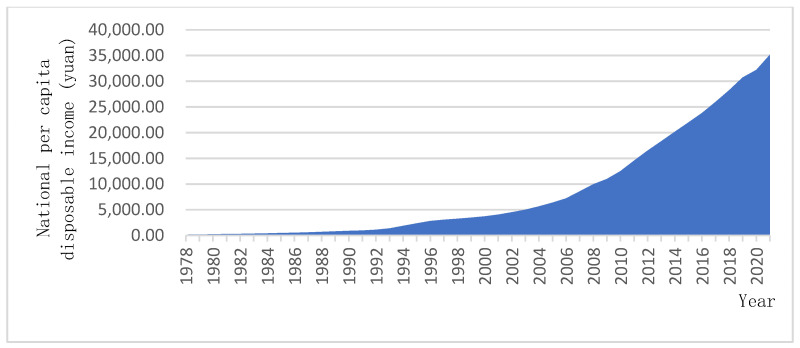
National per capita disposable income since 1978. (Source: Compiled from relevant data published by the National Bureau of Statistics of China in previous years).

**Table 1 ijerph-19-08244-t001:** Analytical framework of external support for elderly care-based social enterprises: the “government-society-family” framework.

External Support Subjects	Support Method
Government	Policy support, financial support, legal support, etc.
Social	Organization support, community platform, etc.
Family	Attitude change, economic support, spiritual comfort, etc... ^1^

^1^ Source: Produced by the author.

**Table 2 ijerph-19-08244-t002:** Profile of interviewees.

Interview Subject Coding (According to the Standard of the First Letter of the Surname+ a Number)	Gender	Category	Total Cumulative Length of Interviews
L1	Female	Founder	14 h
W2	Female	Management	6 h
C3	Male	Management	5 h
Y4	Female	Experts	4 h
L5	Female	Medical and nursing staff	4 h
X6	Male	Expert	4.5 h
W7	Female	Medical and nursing staff	4 h
H8	Female	Caregivers	4 h
T9	Female	Caregivers	5.5 h
W10	Female	Caregivers	5.5 h
L11	Female	Caregivers	4 h
Z12	Female	Family member of older individual	3 h
G13	Female	Family member of older individual	2 h
L14	Male	Family member of older individual	2 h
W15	Male	Family member of older individual	2 h
Z16	Female	Older individual	2 h
Y17	Male	Older individual	2 h
Z18	Female	Older individual	2.5 h

**Table 3 ijerph-19-08244-t003:** Differences in construction subsidies in selected regions (yuan).

Geographical Area	Jiangsu Province Regulation	Suzhou	Nantong	Xuzhou
Per bed	10,000	20,000	10,000	10,000 ^1^

^1^ Source: Based on recent support policies for elderly service providers announced by municipal governments.

**Table 4 ijerph-19-08244-t004:** Differences in operating subsidies in selected regions (yuan).

Geographical Area	Nanjing	Suzhou	Nantong	Xuzhou
Disabled elderly/Month (city domicile)	300	250	60	120
Partially disabled elderly/month (city domicile)	200	150	50	100 ^1^

^1^ Source: Based on recent support policies for elderly service providers announced by municipal governments.

**Table 5 ijerph-19-08244-t005:** Differences in care subsidies in selected areas (yuan).

	Nanjing	Suzhou	Nantong	Xuzhou
Severely disabled elderly/day	--	26	70	30
Moderately disabled elderly/day	--	20	30	-- ^1^

^1^ Source: Based on recent support policies for elderly service providers announced by municipal governments.

**Table 6 ijerph-19-08244-t006:** Pension income levels per household in selected regions, 2018 (yuan).

Region	All Residents	Urban Residents	Rural Residents
Suzhou	7870	9534	4129
Changzhou	7026	9019	2828
Nantong City District	5504	6613	1772
Yangzhou	4521	6935	675
Xuzhou	3496	5477	564
Lianyungang	2905	4396	901 ^1^

^1^ Source: Based on data compiled from the 2019 Statistical Yearbook of six municipalities in Jiangsu Province.

## Data Availability

Jiangsu Statistical Yearbook (2019): http://tj.jiangsu.gov.cn/col/col76362/index.html (accessed on 13 May 2022). National Bureau of statistics of China, National Database: https://data.stats.gov.cn/english/easyquery.htm?cn=C01 (accessed on 13 May 2022).

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
