# Peer review of "External Support for Elderly Care Social Enterprises in China: A Government-Society-Family Framework of Analysis"

_ijerph, 2022, doi:10.3390/ijerph19148244_

Round 1

Reviewer 1 Report

General Concept comments: (weaknesses, method inaccuracies)

Difficult to interpret this final statement in the abstract: Based on this research, the external support system of the elderly care social enterprises is constructed to overcome the constraints of insufficient external support in the sustainable development of the elderly care social enterprises.

Specific Comments: (line numbers, table numbers, figure numbers, pointing out inaccuracies, etc.)

Line 115: is Twitter an appropriate reference for a research article?

Line 128-129: APA or another citing format?? Atieno, R. (2009), Herrington, M., et al. (2009) and Maas, G. Herrington, M. (2007). Need consistency

Line 181, 199, 240, 243: no pub date for author citation (consistency throughout the manuscript)

Line 33 et al: Overuse of the phrase ‘on the one hand’, ‘on the other hand’

Line 206-211; 248-251: good identification of gap

Line 239, 243: do you need the year with these citing’s???? If they are in quotes, does your style involve indicating page numbers along with the year?

Section 3: Methodology

Good identification of method/design, purpose, aim, types of data collection. This is beneficial for the reading audience.

Line 305-306: would like to see a reference to triangulation rather than reliability/validity (quantitative vs. qualitative). The fact that you mention we in the purpose to grasp the views, attitudes, emotions…is a good reference to the case study factors. Operationalizing the constructs for your study was also referenced. Job well done. (line 297-298)

Paragraph starting with line 558: clarify who is talking about Grandpa Lu (start another paragraph with his story)

Some lack of citing (year, page number). Line 366-367. 369-370. 421-425. 647-648.

Line 511: do you need to spell this out before using the abbreviations?

5.2. Limitations

You mention ‘self-blooding’ etc. This is not something I am familiar with. Is this something common in China? May require further explanation for deeper understanding as to limitation.

Appropriateness of References

Many references are over 5 years old, which is consistent with the scope of the study, however, 75 % are from 1979 through 2018. If there are more current studies or references to elder/social enterprises, it would be beneficial to include in the study (give that a gap was identified)

Author Response

Thank you very much for your approval of the paper. We will reply to each of your suggestions below.

General Concept comments:

1.this final statement in the abstract: Based on this research, the external support system of the elderly care social enterprises is constructed to overcome the constraints of insufficient external support in the sustainable development of the elderly care social enterprises. Modified toFinally, an external support system of Chinese elderly care social enterprises is constructed to help more elderly care social enterprises overcome the lack of external support in the development process.

Specific Comments:

2.Line 115: is Twitter an appropriate reference for a research article?: This Twitter is an example from the author. To avoid misleading, delete “Twitter”.

3.Line 128-129: APA or another citing format?? Atieno, R. (2009), Herrington, M., et al. (2009) and Maas, G. Herrington, M. (2007). Need consistency. Line 181, 199, 240, 243: no pub date for author citation (consistency throughout the manuscript): The type of problems that appears in the article has been modified. A total of 15 changes were made.

4.Line 33 et al: Overuse of the phrase ‘on the one hand’, ‘on the other hand’: 'on the one hand' has been removed. or change it to“Firstly,”“Secondly”. A total of 11 changes were made.

5.Line 239, 243: do you need the year with these citing’s???? If they are in quotes, does your style involve indicating page numbers along with the year: The quotation marks are of little use; they have been removed. In order to unify the format year only appears in the last reference.

Section 3: Methodology

6.Line 305-306: would like to see a reference to triangulation rather than reliability/validity (quantitative vs. qualitative).

Answer: By collecting data from professional authorities and other social enterprises involved in elderly care services, we further confirmed the typicality of GY in this type of social enterprises. We spent six months participating in observation and field interviews, and recorded first-hand information in real time. In general, the triangulation method in qualitative research strategy was adopted, including the methodological triangulation method and the researcher triangulation method. This approach allowed us to obtain a more comprehensive understanding of the research phenomenon. According to the academic research norms, the names of organizations and personnel in the text have been technically processed (Table 2).

7.Paragraph starting with line 558: clarify who is talking about Grandpa Lu (start another paragraph with his story). For the revision of this section, a new paragraph was created as requested and the narrator of the story was added: In our study of GY, We heard a story about Grandpa Lu from the interviewee W7.

8.Some lack of citing (year, page number). Line 366-367. 369-370. 421-425. 647-648.

Answer: Missing related elements were added

Line 366-367. 369-370. in China at that time, the perception of social enterprise was very vague. On June 3, 2011, the ninth plenary session of the 10th Beijing Municipal Committee of the Communist Party of China adopted the “Opinions of the Beijing Municipal Committee of the Communist Party of China on Strengthening and Innovating Social Management to Promote Social Construction in an All-round Way”, the third part of which encourages localities to “actively support the development of social enterprises and vigorously develop social service industry”. This was the first mention of social enterprises in official documents. On November 11, 2016, the General Office of the Communist Party of China Beijing Municipal Committee Beijing Municipal People's Government issued “Beijing’s 13th Five-Year period of social governance planning”, in which the third part of the fourth article clearly points out that one should “vigorously promote the development of social enterprises focusing on serving the people’s livelihood and carrying out public welfare”. On 23 April 2018, the Chengdu Municipal Government promulgated the “Opinion on Cultivating Social Enterprises for Community Development and Governance”, which was the first relevant policy specifically targeting social enterprises. Thus far, this is the only document specifically aimed at social enterprises; there is no similar national document or guidance at the specific practical level.

Line 421-425"When your legs are an inch thinner, the elderly care service organization will be successful"(It means that the procedure to successfully register an elderly care social enterprise is very cumbersome. Managers need to go to many departments in different regions or different functions to complete the procedures.). (Interview record: 20191023).

Line 647-648: For example, the government can implement the policy of "new homes with new methods and old homes with old methods" when issuing fire certificates for elderly care services, and new elderly care social enterprises should strictly follow the policy when applying for fire certificates. This part is the author's suggestion, there is no time for reference

9.Line 511: do you need to spell this out before using the abbreviations? The modification is to add a full name for the relevant abbreviations:

OECD (Organization for Economic Co-operation and Development); UK(The United Kingdom of Great Britain and Northern Ireland)

5.2. Limitations

10.You mention ‘self-blooding’ etc. This is not something I am familiar with. Is this something common in China? May require further explanation for deeper understanding as to limitation. Added notes on "self-blooding" and "blood transfusion": The statements of “self-blooding” and “external blood transfusion” are widely used in many fields in China, such as the development of public welfare organizations, poverty alleviation and rural revitalization. The meaning of ' self- blooding' is to focus on the internal motivation of things, that is, through their own efforts to achieve development; blood transfusion is an external factor, that is, the development of things through external support.

Appropriateness of References

11.Many references are over 5 years old, which is consistent with the scope of the study, however, 75 % are from 1979 through 2018. If there are more current studies or references to elder/social enterprises, it would be beneficial to include in the study

The following literature is replaced by the latest ones.

[2] Hirschmann, M.; Moritz, A.; Block, J. H. Motives, Supporting Activities, and Selection Criteria of Social Impact Incubators: An Experimental Conjoint Study. Nonprofit and Voluntary Sector Quarterly, 2021,28:1-39.

[15] Li, W. R.; He, X. H. A Brief Analysis on the Tax Policies for Encouraging Social Capital to Participate in the Elderly Service Industry, taxation research. 2021, (12):23-27.

[16] Yin, F.; Zhang, Y. M.; Liu, M.; Li, J. From the Perspective of Supply-side Structure, the Development Constraints and Strategic Choices of Pension Industry in China, Urban Development Studies. 2020,27(04):1-5.

[17] Wu, H. L. The Public Interest Logic and Operation Dilemma of Social Enterprises Providing Pension Service. Journal of Fujian Normal University (Philosophy and Social Sciences Edition), 2017, (01):57-67.

[25] Hung, C.; Wang, L. Institutional Constraints, Market Competition, and Revenue Strategies Evidence from Canadian, Social Enterprises.2021, 32(1):165–177.

[26] Saebi, T.; Foss, N.; Linder, S. Social Entrepreneurship Research: Past Achievements and Future Promises. J. Manag. 2019, 45: 70–95.

[27] Andrea, R. M.; Millán, D. F. Pilar, A. M. The determinants of social sustainability in work integration social enterprises: the effect of entrepreneurship, Economic Research-Ekonomska Istraživanja, 2021,34(1):929-947.

[28] De Ruysscher, C.; Claes, C.; Lee, T. et al. A Systems Approach to Social Entrepreneurship. International Journal of Voluntary & Nonprofit Organizations, 2017,28: 2530–2545.

[29] Czischke, D.; Gruis, V.; Mullins, D. Conceptualising social enterprise in housing organisations. Housing Studies. 2012,27(4),418-437

[30] Davies, I. A.; Haugh, H.; Chambers, L. Barriers to Social Enterprise Growth. Journal of Small Business Management. 2019,57(4):1616-1636

[31] Varendh-Mansson, C.; Wry, T.; Szafarz, A. Anchors Aweigh? Then Time to Head Upstream: Why We Need to Theorize “Mission” Before “Drift”. Academy of Management Review. 2020,45(1),230-234.

[32] Islam, M.N.; Ozuem, W.; Bowen, G.; Willis, M.; Ng, R. An Empirical Investigation and Conceptual Model of Perceptions, Support, and Barriers to Marketing in Social Enterprises in Bangladesh. Sustainability 2021, 13, 345.

[33] Choi, D.; Park, J. Local government as a catalyst for promoting social enterprise. Public Management Review,2021,23(5), 665-686.

[34] Mazzei, M.; Steiner, A "What about efficiency? Exploring perceptions of current social enterprise support provision in Scotland." Geoforum,2021,118:38-46.

[6] Jf, A.; Pk, B., Bh, C. et al. Using micro-geography to understand the realisation of wellbeing: A qualitative GIS study of three social enterprises. Health & Place, 2020, 62: 102293.(替换文献6)

[43] Powell, M.G., Gillett, A. and Doherty, B. Sustainability in social enterprise: hybrid organizing in public services. Public Management Review, 2019,21 (2):159-186.(替换文献43)

[80] Phillips, W.; Alexander, EA.;Lee, H.Going It Alone Won't Work! The Relational Imperative for Social Innovation in Social Enterprises. Journal of business ethics, 2019,156(2):315-331.

[81]原来的[80]修改为[81]

[82] Kiss, J.; Krátki, N.; Deme, G. Interaction between social enterprises and key actors shaping the field: experiences from the social and health sectors in Hungary. Social Enterprise Journal, 2021, 17(4):625-646.

[83] Andrea, R. M.; Millán, D. F.; Pilar, A. M. The determinants of social sustainability in work integration social enterprises: the effect of entrepreneurship. Ekonomska Istraživanja/Economic Research, 2020,34(2):1-19.

[84] Fiona, H.; Artur, S.; Micaela, M.; Catherine, D. Social enterprises' impact on older people's health and wellbeing: exploring Scottish experiences. Health Promotion International,2020,35:1074-1084.

[85] Hoyos, A.; Angel-Urdinola, D. F.Assessing international organizations' support to social enterprise. Development Policy Review, 2019,37:213-229.

[86] Powell, M.G., Gillett, A. and Doherty, B. Sustainability in social enterprise: hybrid organizing in public services. Public Management Review, 2019,21 (2):159-186.

[87] Wu, Z. X. Research on the Legality Acquisition Mechanism of Social Enterprises from the Perspective of Social Interaction. Frontiers in Economics and Management,2021,2(7):152-165.

[88] Hirschmann, M.; Moritz, A.;Block, J. H. Motives, Supporting Activities, and Selection Criteria of Social Impact Incubators: An Experimental Conjoint Study. Nonprofit and Voluntary Sector Quarterly,2021. https://doi.org/10.1177/08997640211057402

[89] Jianyuan, H.; Yaqing CH. Is the function of family pension weakened--Based on the dual investigation of economy and service. Social security review.2020,4(02):131-145. (Chinese)

[90]Yakita A. Economic development and long-term care provision by families, markets and the state. The Journal of the Economics of Ageing, 2020, 15.

Thank you again for your comments and suggestions. We have responded to your suggestions. I hope you are satisfied.

Reviewer 2 Report

I am honored to have had the opportunity to read such a good study.

This study is a good research topic as it is an important issue to identify the significance of each area of ​​social enterprises.

Just a few comments.

1. The meaning of great heterogeneity is not clear in the sentence “Social inequality and the wide range of fields and populations involved in social problems have led to great heterogeneity among social enterprises” in the introduction. If it is written for the logic that it is necessary to check each area because it is different, it should be described more clearly.

2. Among social enterprises, elderly care social enterprises have lower sustainability and were they selected as the subject of this study because of concerns? However, in this text, evidence is needed to support the logic of the researchers that the elderly care area is more vulnerable than other areas or requires more external support.

3. The difference between external support and financial independence should be further described to enhance readers' understanding. For example, if financial independence, which has already been dealt with in previous studies, has been revealed in a narrow sense, the meaning of external support should be emphasized more.

4. 2. From the contents of the Literature Review section, I could see the efforts of the authors. However, it is suggested to modify the paragraphs as a whole to better reveal the purpose of the study.

What should be dealt with first is how the researchers defined the external support of social enterprises in the care of the elderly and to what extent they were dealt with in existing studies. It is also the basis for which the most important framework of this study, the analysis framework, was derived.

5. The reason for choosing GY as a typical case is described. However, it is insufficient for GY to be defined as a typical type due to reasons such as that it has been developing for 24 years. The function as a social enterprise of care for the elderly should be described.

6. 4.1.1. Most of the contents of the Lack of social enterprise legality policy are related to the background or discussion of the study. Therefore, it is difficult to see only the results of the studies that the studies have drawn. Therefore, it is recommended to move this content to the front part.

Also, the results section contains a lot of discussion content, so it needs to be rewritten.

Author Response

Author's Reply to the Review Report (Reviewer 2)

Thank you very much for your approval of the paper. We will reply to each of your suggestions below.

  1. The meaning of great heterogeneity is not clear in the sentence “Social inequality and the wide range of fields and populations involved in social problems have led to great heterogeneity among social enterprises” in the introduction. If it is written for the logic that it is necessary to check each area because it is different, it should be described more clearly.

A new explanation was added as follows: In China, for example, social enterprise services cover 16 social areas, including environmental protection, accessibility services, community development, public finance, pension, education, employment of vulnerable groups, agriculture, poverty alleviation, the Internet, public security, women's rights and interests, and focus on 14 specific groups. There are great differences among the social enterprises services available in different social fields and groups.

  1. Among social enterprises, elderly care social enterprises have lower sustainability and were they selected as the subject of this study because of concerns? However, in this text, evidence is needed to support the logic of the researchers that the elderly care area is more vulnerable than other areas or requires more external support.

A new argument was added to solve the problem: In China, with the deepening of population aging, elderly care faces a series of challenges. The sector’s single dependence on the government has been unable to meet the growing demand for pension services, and the enthusiasm of social capital to enter the pension service industry needs to be improved [15]. The overall pension supply has shown a low level of involution [16]. As the main body of pension supply, pension-oriented social enterprises are still in the exploratory stage and face difficulties such as concepts, laws, practices and talents [17]; thus, they lack the impetus for sustainable development.

[15] Li, W. R.; He, X. H. A Brief Analysis on the Tax Policies for Encouraging Social Capital to Participate in the Elderly Service Industry, taxation research. 2021, (12):23-27.

[16] Yin, F.; Zhang, Y. M.; Liu, M.; Li, J. From the Perspective of Supply-side Structure, the Development Constraints and Strategic Choices of Pension Industry in China, Urban Development Studies. 2020,27(04):1-5

[17] Wu, H. L. The Public Interest Logic and Operation Dilemma of Social Enterprises Providing Pension Service. Journal of Fujian Normal University (Philosophy and Social Sciences Edition), 2017, (01):57-67.

  1. The difference between external support and financial independence should be further described to enhance readers' understanding. For example, if financial independence, which has already been dealt with in previous studies, has been revealed in a narrow sense, the meaning of external support should be emphasized more.

This problem is explained in lines 52-57 of the paper, as follows:

However, most of them have focused on the internal capacity building of social enterprises themselves, such as the financial independence [21], legitimacy [22], and competitiveness [23] of social enterprises. Fewer studies have been conducted from the perspective of external support of organizations. The sustainable development of any organization cannot be achieved without external support; therefore, our research focuses on the external support of social enterprises.

  1. 2. From the contents of the Literature Review section, it is suggested to modify the paragraphs as a whole to better reveal the purpose of the study.

What should be dealt with first is how the researchers defined the external support of social enterprises in the care of the elderly and to what extent they were dealt with in existing studies. It is also the basis for which the most important framework of this study, the analysis framework, was derived.

Answer: The content of the literature review part, especially the first part and the second part have been rewritten, which can be clearly reflected in the annex.

5.The reason for choosing GY as a typical case is described. However, it is insufficient for GY to be defined as a typical type due to reasons such as that it has been developing for 24 years. The function as a social enterprise of care for the elderly should be described.

Answer: Three criteria were used to select GY as a typical case: social enterprise, the provision of elderly care services, and the extension of development time. In addition, the paper adds an overview of GY to give the reader a more comprehensive understanding of GY. These changes can be well reflected in the annex.

  1. 4.1.1. Most of the contents of the Lack of social enterprise legality policy are related to the background or discussion of the study. Therefore, it is difficult to see only the results of the studies that the studies have drawn. Therefore, it is recommended to move this content to the front part.

Answer: 4.4.1 is recommended to move this content to the front part. (3.3)

7.the results section contains a lot of discussion content, so it needs to be rewritten.

Answer: The section of results, conclusions and discussion were changed. The changes to this section are reflected in the attachment. Please download and view.

Thank you again for your comments and suggestions. We have responded to your suggestions. I hope you are satisfied.

Reviewer 3 Report

At the outset, the reviewer points out that the layout of the work is incorrect, which is inconsistent with the publishing standard of the MDPI (IJERPH). There is a lack of thesis and aims, additional points not included in the template must be removed, and the theoretical part itself needs to be shortened. The entire point 2 "literature review" is, in the opinion of the reviewer, unnecessary, because it is the basis for the research, a conceptual matrix for the research design, and is not applicable in the article. It also makes it too long and incomprehensible. In the opinion of the reviewer, presenting numerous theories at this stage, only some of which are applicable in the research is unnecessary.

Work in point 2 resembles a textbook, not an article in which a specific research goal is pursued.

At no point in the study was indicated what the aim of the study was, or what research theses were correlated with the aim (or with complementary aims). Thus, the work is not of a scientific nature.

The authors indicate that the research methodology was based on a case study, but it was not presented anywhere in this case. It is not known what institution it was, how it was organized, what was its business profile, or what was the market environment. In the opinion of the reviewer, therefore, this is not a qualitative case study analysis, but a description of selected criteria of an institution that is not known to us. The selection of the inclusion criteria (section 3.2) indicated a number of criteria, but they are very general and it is not known how many units (institutions) were subject to these criteria (at each stage of the selection, how many met the criteria and how many were excluded).

The fourth point, "results", is in fact a theoretical description of the geopolitical situation related to China's social policy, and could well be included in the theoretical introduction as well as in the discussion. The results of the case study are not presented in this part, anyway, it is not known what the course of the analysis is, or what activities within the framework of the case study were carried out.

Part 5 "conclusions and discussion" is incorrectly connected. The results of the study are one thing, and the discussion (of these results with the literature or with real conditions) is another. The authors combined these points, and in the content, they carried out another desktop research analysis (discussion of the geopolitical situation).

In the part concerning limitations, the limitations of this study were not really included, but once again we can see a general statement about the subject of senior care. Thus, I consider the content of this point incorrect.

The overall assessment of the work is negative. The reviewer does not see any attributes of scientific research in the work. It is true that a very extensive desk-research analysis was carried out, based on numerous literature sources, but it is not known why. What is the aim of the research, what theses have been defined, for what purpose the research was carried out and in which unit the case study was carried out? All these questions mean that, in the opinion of the reviewer, the work should be rejected from publication or requires complete rebuilding.

Author Response

Author's Reply to the Review Report (Reviewer 3)

Thank you very much for your approval of the paper. We will reply to each of your suggestions below.

  1. At the outset, the reviewer points out that the layout of the work is incorrect, which is inconsistent with the publishing standard of the MDPI (IJERPH). There is a lack of thesis and aims, additional points not included in the template must be removed, and the theoretical part itself needs to be shortened.

Answer: the layout of the work has been modified accordingly. But the literature review is preserved. Significant changes have been made to this section. Please see the attachment of the specific changes, thank you very much.

  1. The entire point 2 "literature review" is, in the opinion of the reviewer, unnecessary, because it is the basis for the research, a conceptual matrix for the research design, and is not applicable in the article. It also makes it too long and incomprehensible. In the opinion of the reviewer, presenting numerous theories at this stage, only some of which are applicable in the research is unnecessary.

Answer: Based on your suggestion, we revised the first half of the literature review of the paper, but retained the forming part of the analytical framework. We believe that this paper needs the existence of literature reviews, especially the forming part of the analytical framework, because it is the basis of the analysis of the results and discussions in this paper.

  1. Work in point 2 resembles a textbook, not an article in which a specific research goal is pursued.

At no point in the study was indicated what the aim of the study was, or what research theses were correlated with the aim (or with complementary aims). Thus, the work is not of a scientific nature.

Answer: We added the research purpose of our paper to line 337 - 341.

  1. The authors indicate that the research methodology was based on a case study, but it was not presented anywhere in this case. It is not known what institution it was, how it was organized, what was its business profile, or what was the market environment. In the opinion of the reviewer, therefore, this is not a qualitative case study analysis, but a description of selected criteria of an institution that is not known to us. The selection of the inclusion criteria (section 3.2) indicated a number of criteria, but they are very general and it is not known how many units (institutions) were subject to these criteria (at each stage of the selection, how many met the criteria and how many were excluded).

Answer:  Thank you for your suggestion. We have made a comprehensive revision of the 3. Methodology and added a detailed description of the case. Please see the specific changes in the 3. Methodology.

  1. The fourth point, "results", is in fact a theoretical description of the geopolitical situation related to China's social policy, and could well be included in the theoretical introduction as well as in the discussion. The results of the case study are not presented in this part, anyway, it is not known what the course of the analysis is, or what activities within the framework of the case study were carried out.

Part 5 "conclusions and discussion" is incorrectly connected. The results of the study are one thing, and the discussion (of these results with the literature or with real conditions) is another. The authors combined these points, and in the content, they carried out another desktop research analysis (discussion of the geopolitical situation).

In the part concerning limitations, the limitations of this study were not really included, but once again we can see a general statement about the subject of senior care. Thus, I consider the content of this point incorrect.

Answer: "results", "conclusions and discussion" have been revised. Some of the contents in “results” are deleted and modified, “conclusions and discussions” was divided into two parts and revised a lot. Please see the specific revisions to line 515-1096.

The overall assessment of the work is negative. The reviewer does not see any attributes of scientific research in the work. It is true that a very extensive desk-research analysis was carried out, based on numerous literature sources, but it is not known why. What is the aim of the research, what theses have been defined, for what purpose the research was carried out and in which unit the case study was carried out? All these questions mean that, in the opinion of the reviewer, the work should be rejected from publication or requires complete rebuilding.

Thank you again for your comments and suggestions. We have responded to your suggestions. I hope you are satisfied.

Reviewer 4 Report

The article presents a case study of an older people social enterprise in China (GY) used as a starting point for a critical discussion of the role of social support and external support such social enterprise would need from the government and the society as a whole. The authors argue for the need to get community (support) in assuring the continuation of social  enterprises of older adults in China and also the importance of  the adequacy of public policies and  government coherence in  measures targeting  social  enterprises.
The article would benefit from some improvements and language adjustments

·       Avoid stereotypical language and the use of “elderly”. Instead use older people, older persons or similar non-judgmental terms

·       What is an older people care social enterprise should be   better clarify in the text. Also what would be the particularities of such social enterprise compared with other types of social  enterprises

·       What is an external support in the context of the manuscript? What an internal support will be? What is the role of an external support in a social enterprise in general? What about the care social enterprise?

·       Provide arguments for choosing GY as a case for study. Why is this case typical for the topic addressed in the manuscript?

·       Provide a separate section of conclusions and a separate section of discussion

Author Response

Author's Reply to the Review Report (Reviewer 4)

Thank you very much for your approval of the paper. We will reply to each of your suggestions below.

  1. Avoid stereotypical language and the use of “elderly”. Instead use older people, older persons or similar non-judgmental terms

Answer: The language using stereotypes and the “elderly” in the paper have been revised. They can be reflected in the attached articles.

  1. What is an older people care social enterprise should be   better clarify in the text. Also what would be the particularities of such social enterprise compared with other types of social  enterprises

Answer: In part 3.2, clarification of the elderly care social enterprise: Second, the social enterprises examined in the current study provide elderly care services; thus, other types of social enterprises are not within the scope of this study. There are many fields of social enterprise services, such as those pertaining to children and youth, barrier-free services, rural development, ecological protection, elderly care and so on. Given that there are differences between the different fields of social enterprises, we only focus on the field of elderly care of social enterprises. These enterprises will be collectively referred to as elderly care-oriented social enterprises. Their service objects are mainly elderly individuals and their families, and their function is mainly to provide various long-term or short-term old-age care services, such as life care, medical care, spiritual comfort, emergency relief, breathing services, home services and so on.

  1. What is an external support in the context of the manuscript? What an internal support will be? What is the role of an external support in a social enterprise in general? What about the care social enterprise?

Answer: In order to better reflect the external support and internal support, the first and second parts of the literature were revised.

Research on social enterprise support

A social enterprise is a business activity that is driven by a social mission [24-25], which does not mean that the enterprise itself is sustainable; it needs to maintain its development in a certain way. This approach can be seen as both the ability of social enterprises to engage in self-reliance (an internal support) and various forms of external support. The first aspect currently garners more academic attention as it consists of more aspects, including leadership, stakeholders, resilient development, sharing practices, brand equity, finance, market competitiveness, social entrepreneurship and other factors [6,18,26,27]. Social enterprises emphasize the market competitiveness of products and services. For example, the overall goal of BOSKE (i.e., BOSKE Bakery Cafe´, which is located in Taiwan and is a social services organization that provides community living and vocational rehabilitation services to adults with intellectual and developmental disabilities) is for people to buy its products because they enjoy its baked goods and services rather than because they feel pity, compassion, or responsibility toward the individuals [28].

The second aspect is relatively weak in regard to garnering people’s attention. Funding sources have a significant impact on the survival and development of social enterprises, thereby making them more dependent on government grants and donor funds, especially in the early stages of creation [29]. Institutional constraints make social enterprises rely more on commercial income during their development [25,30]. This makes it extremely easy for a social enterprise to engage in goal task drift and become faced with the dilemma of social enterprise dual goals [31]. Some scholars believe that infrastructure is an important obstacle to the survival and development of social enterprises [32]. Faced with these obstacles, the local government's support to social enterprises is very effective [33]. However, when the government provides support, it is temporary and lacks a comprehensive method guidance. This makes such support inefficient and wasteful [34.]

Jian, et al. used a content analysis approach to study health organizations seeking social support, including informational, emotional, and instrumental support, through social networking sites [35]. In Berlin, IQ Consulting is a social innovation agency and think tank that supports social enterprise projects with services such as planning, coordination, training and evaluation [36]. IKure was founded in 2010 with incubation support from the Indian Institute of Technology Kharagpur and Webel Venture Fund, an early-stage incubator established by the West Bengal Government Corporation [37]. Upeeffect was founded in 2016 by social entrepreneurs Sheeza Shah and Sharjeel Chaudhry and is based in London; it currently serves social entrepreneurs in 14 countries, and its crowdfunding mentoring model has resulted in a 95% campaign success rate for social entrepreneurs [38]. Upeeffect CEO Shah attributed the low success rate of social enterprise campaigns to a lack of campaign support. The Frugal Innovation Lab at Santa Clara University helps social enterprises by using existing tools [39] to achieve a balance between their social mission and financial aspects. Adequate financial support is an important aspect for the sustainability of social enterprises [40], and the lack of financing is one of the main constraints faced by entrepreneurs when starting a business [41], as shown by the studies of Atieno, Herrington et al., and Maas [42-43]. Herrington reported that in the absence of financial support, entrepreneurs cannot predict the survival and growth of their projects [44], and Casson pointed out that inadequate capital structure and lack of financial resources are the main reasons for the unsustainability of social enterprises [45]. Martin and Eisenhardt used resource-based theory to argue for the financing needs of entrepreneurs, arguing that entrepreneurs need resources such as fixed assets and working capital to gain a competitive advantage in the market [46].

In short, the existing literature on social enterprise support research is still quite weak because it mainly focuses on how to maintain the balance between a social mission and its financing with single support, such as local government support; i.e., the research lacks a comprehensive discussion of support systems. We believe that it is not enough that the support needed for the sustainable development of social enterprises only comes from financial support. It also needs more support from other aspects, and it is better to form a support system.

  1. Provide arguments for choosing GY as a case for study. Why is this case typical for the topic addressed in the manuscript?

Answer: In addition to presenting the criteria for case selection, the methodology section adds an overview of the case (GY) to better illustrate its representativeness and typicality.

case profile

GY was founded in 1999 and has been in development for more than 20 years. In 1999, the government began to allow social capital to enter the field of elderly care. The first private elderly care institutions in our country were established in 1999; the current Chinese population ageing problem also began in this year. Prior to 1999, care services for the elderly population were basically carried out by the children in a family; thus, the demand for socialization of care services for the elderly population was not strong. At this point, China's elderly care services did not cause widespread concern within the community or serve as a social problem. As the country entered the current era of population ageing, families began to increasingly struggle to provide care for their elderly relatives. The rigid demand of China's elderly care services was seriously insufficient, which caused widespread concern from all walks of life. In this context, the elderly care industry began to develop.

In its initial period, GY was a simple private enterprise for the aged, which mainly provided life care services for the aged with disabilities and mental handicaps and cooperated with hospitals to provide medical rehabilitation services. During this period, GY had 60 beds and offered the concept of integrated care. At that time, due to the lack of a corresponding policy system and the support of relevant departments, the implementation of integrated care concept was very difficult. In 2003, GY was transformed into a public-funded private pension enterprise that was supported by government infrastructure; the number of beds increased to 106.

In 2011, under the guidance of national policy, GY began to explore the chain operation model, and successfully created a second retirement home. Thus far, there are 12 chain organizations. The chain’s geographical area covers the southern, northern and central areas of Jiangsu Province. At the time of the opening of the second institution, GY put forward its social value concept of providing high-quality services for elderly and disabled individuals at low cost; these services included personal health care, diet and personal care, home health care and social and recreational activities. GY also regularly participated in the community to carry out free health check-up activities, thereby going as far as possible to solve the problem of unattended elderly individuals. Thus, GY had begun to pursue the dual goals of social enterprise.

However, in China at that time, the perception of social enterprise was very vague. On June 3, 2011, the ninth plenary session of the 10th Beijing Municipal Committee of the Communist Party of China adopted the “Opinions of the Beijing Municipal Committee of the Communist Party of China on Strengthening and Innovating Social Management to Promote Social Construction in an All-round Way”, the third part of which encourages localities to “actively support the development of social enterprises and vigorously develop social service industry”. This was the first mention of social enterprises in official documents. On November 11, 2016, the General Office of the Communist Party of China Beijing Municipal Committee Beijing Municipal People's Government issued “Beijing’s 13th Five-Year period of social governance planning”, in which the third part of the fourth article clearly points out that one should “vigorously promote the development of social enterprises focusing on serving the people’s livelihood and carrying out public welfare”. On 23 April 2018, the Chengdu Municipal Government promulgated the “Opinion on Cultivating Social Enterprises for Community Development and Governance”, which was the first relevant policy specifically targeting social enterprises. Thus far, this is the only document specifically aimed at social enterprises; there is no similar national document or guidance at the specific practical level.

There are many geriatric care social enterprises, or potential social enterprises, in the field of practice that are similar to GY. The main reason for calling GY a potential social enterprise is that the status of their social enterprise is not clear. Although GY has been practicing the dual goals of social enterprises, they have no clear identity at the legal level, which results in its development process being subject to many restrictions. Initially, GY was positioned as a private elderly care institution. Later, it cooperated with the government and became a public-private enterprise that was supported by the government. At present, it is positioned as an elderly care social enterprise. However, there are some controversies over elderly care social enterprises in both academic and practice fields, and the dual objectives of social enterprises (social and economic benefits) are like a double-edged sword because these enterprises are considered both a model innovation and a source of controversy. In short, social enterprises started late in China and are still in the exploratory stage. Many private nongovernment enterprises, institutions or organizations already have various elements or characteristics of social enterprises but do not have the "self-awareness" of social enterprises. Many social enterprises are torn between presenting themselves as general commercial enterprises or nonprofit organizations for the public good, and they do not have a clear understanding of their own function. "When I first came to GY, I thought it was a nonprofit social organization but later found that GY was in the development process, and except for government-related preferential policies, its development basically relies on its own profitability. However, there are a lot of elderly care institutions in society; no matter public, private or public-private construction, their identities are difficult to accurately position, so the positioning of GY is really difficult to say” (Interview record: 20201018).

  1. Provide a separate section of conclusions and a separate section of discussion

Answer: The conclusion section has been separated from the discussion section based on your suggestion. The discussion section has also been revised. Please see the relevant changes in the attachment.

Thank you again for your comments and suggestions. We have responded to your suggestions. I hope you are satisfied.

Reviewer 5 Report

This paper is interesting and informative. It highlights the differences in handling emerging issues relating to an aging population and setting up social enterprises – particularly within the challenging context of China.  It makes a meaningful contribution by defining the government-society-family framework.

Overall, I have virtually no issues with the content. It is a straightforward descriptive case study intermingled with a policy review.  Essentially, the example is used to illustrate what has occurred, what is happening, and what still needs to be done with respect to social enterprises for a growing older population in China.

There were some minor content questions.  The term “potential eldercare” to describe “GY” is misleading (Lines 284-285). Initially, I thought this was a hypothetical social enterprise when I first read it. It becomes clear that “GY” is an actual enterprise and this is a case study. Some clarity is needed here.  

I did not understand the quotation “When your legs are an inch thinner, the elderly care service organization will be successful” from Interview Record 20191018 (Line 422ff).  I am not sure that that means; it needs to be explained – especially if it is a cultural idiom that is not (well) known outside of China.

Finally, there is some confusion about money. Both Renminbi and Yuan are used.  The discussion starts with the common currency with RMB (the abbreviation for Renminbi) being used both before and after amounts of money (Line 451ff). Then three tables (Tables 3, 4, and 5) have no monetary denomination listed. After that, Table 6 and the subsequent paragraphs uses “Yuan” – which may be technically correct since CNY is generally used for an account of China’s economic and financial system discussions. Even if these abbreviations and terms are used correctly, which is possible given the context, the changes and inconsistencies may be confusing to a reader from elsewhere.

Mostly though, I found minor stylistic and structural issues. While the paper is generally well-written, occasionally the text is overly verbose or dense.  The best example of this is the first sentence of the abstract (Line 1) which is awkward and features the phrase “global aging of population.”  What is meant by the term “BOSKE” (Line 96) is unclear. The “GY” indicator for the case study social enterprise is used twice in-a-row (at the end of one sentence and the beginning of the next sentence) (Line 360).

The author listings for in-text references in the literature review only need to include the last names. Currently, the last names are followed by the (unnecessary) first initial. (The first instance appears on Line 113 – but it is found throughout the paper). The listings appear to be pulled from the references where the last name followed by initial style is appropriate

Finally, the construction of the interview identifiers in Table 2 is both inexact (as “ZH” is not the first letter while it is the first sound) and overly detailed (I do not think it is necessary to note that it is a capital letter).  So, the text could use some adjustment there.

Overall, these are minor issues that should be able to be easily addressed.

Author Response

Thank you very much for your approval of the paper. We will reply to each of your suggestions below.

  1. About The term “potential eldercare” to describe “GY” is misleading (Lines 284-285). Modified to: For this article, we adopted a qualitative case study approach and selected a potential elderly care social enterprise ("potential social enterprises" can also be understood as "quasi-social enterprises", i.e., a phenomenon that occurs in the current environment of China's social enterprise certification system, where there are physical organizations that are not officially recognized),
  2. About the explanation of “When your legs are an inch thinner, the elderly care service organization will be successful” from Interview Record 20191018 (Line 422ff). Modified to: "When your legs are an inch thinner, the elderly care service organization will be successful". This means that the procedure to successfully register an elderly care social enterprise is very cumbersome, i.e., managers need to go to many departments in different regions or different functions to complete the necessary procedures.
  3. Amendments on some confusion about money.

Answer: The authors have revised all references to RMB and Yuan in the text to Yuan. List the monetary denominations of the three tables (Tables 3, 4, and 5).

  1. About the text overly verbose or dense modification:

(1)Revision of the summary section: As Chinese population ageing becomes increasingly severe, the contradiction between supply and demand in pension services is becoming increasingly serious. The development of elderly care social enterprises plays an important role in solving this contradiction. Such development comes from both the enterprise's own capacity building and from external support. There are abundant studies on the capacity-building of pension social enterprises in the existing literature, but there are relatively few studies on their external support. In order to better study the external support of elderly care social enterprises in China, we adopted the case study method, we selected GY (a typical elderly care social enterprise in China) as a case study according to certain criteria, and we conducted a series of discussions. First, an analytical government-society-family framework is constructed. Second, it is argued that there is insufficient external support for elderly care social enterprises. At the government level, there is a lack of policies, difficulties in implementation and significant geographical differences; at the social level, there are weak support platforms and a lack of community supports; and at the family level, there are constraints in regard to traditional concepts and the ability to pay. Finally, an external support system of Chinese elderly care social enterprises is constructed to help more elderly care social enterprises overcome the lack of external support in the development process.

(2)the term “BOSKE” (Line 96) (i.e., BOSKE Bakery Cafe´, which is located in Taiwan and is a social services organization that provides community living and vocational rehabilitation services to adults with intellectual and developmental disabilities)

(3)The “GY” indicator for the case study social enterprise is used twice in-a-row (at the end of one sentence and the beginning of the next sentence) (Line 360). Modify ”GY” to “It”.

  1. According to the standard of the author listings for in-text references in the literature review only need to include the last names.

Answer: Make changes to areas that do not meet the standard. There are 15 modifications.

  1. the construction of the interview identifiers in Table 2 is both inexact.

Modified to: According to the standard of the first letter of the surname+ number

Thank you again for your comments and suggestions. We have responded to your suggestions. I hope you are satisfied.

Round 2

Reviewer 3 Report

I have no additional comments, the work has been improved as suggested. I do not submit any other comments. I accept this work for publish

This manuscript is a resubmission of an earlier submission. The following is a list of the peer review reports and author responses from that submission.